The soft tissue and skeletal anatomy of two Late Jurassic ichthyosaur specimens from the Solnhofen archipelago

http://orcid.org/0000-0002-6806-1411 Delsett Lene L. 1 2 DelsettL@si.edu
http://orcid.org/0000-0001-5700-8730 Friis Henrik 3
Kölbl-Ebert Martina 4
Hurum Jørn H. 3
1 Department of Paleobiology, National Museum of Natural History, Smithsonian Institution , Washington, District of Columbia , United States of America
2 Centre for Ecological and Evolutionary Synthesis, University of Oslo , Oslo , Norway
3 Natural History Museum, University of Oslo , Oslo , Norway
4 Department of Earth and Environmental Sciences, Ludwig Maximilian Universität , Munich , Germany
Noto Christopher
Electronic publication date: 2022 Apr 7
Publication date: 2022
Volume: 10
Electronic Location ID: e13173
Received 2021 Nov 26; Accepted 2022 Mar 6
Copyright: © 2022 Delsett et al.
Copyright year: 2022
Copyright holder: Delsett et al.
License: This is an open access article distributed under the terms of the Creative Commons Attribution License, which permits unrestricted use, distribution, reproduction and adaptation in any medium and for any purpose provided that it is properly attributed. For attribution, the original author(s), title, publication source (PeerJ) and either DOI or URL of the article must be cited.
License URL: https://creativecommons.org/licenses/by/4.0/

Keywords: Ichthyosaur, Solnhofen, Taphonomy, Soft tissue, Aegirosaurus, Phosphatization, Ophthalmosauridae

Funding: Hans and Helga Reusch grant to Lene Liebe Delsett Fulbright research grant to Lene Liebe Delsett This work was supported by a Fulbright research grant and a Hans and Helga Reusch grant to Lene Liebe Delsett. The funders had no role in study design, data collection and analysis, decision to publish, or preparation of the manuscript.

==============================
Ichthyosaurs from the Solnhofen Lagerstätte are among the only examples of soft tissue preservation in the major Middle Jurassic–middle Cretaceous family Ophthalmosauridae. However, few such specimens are currently described, and the taphonomical pathways for the preservation of soft tissue are not well understood. In order to answer this, two new ichthyosaur specimens, one nearly complete and one isolated tail, are described here. The nearly complete specimen is assigned to Aegirosaurus sp. It is accompanied by large amounts of incrustation pseudomorphs (epimorphs) of soft tissue preserved as apatite. It also preserves a nearly complete gastral basket, for the first time in ophthalmosaurids. Soft tissue samples were analyzed using X-ray diffraction (XRD) and scanning electron microscopy (SEM) coupled with energy dispersive spectroscopy (EDS) analysis. The analyses confirm the presence of apatite, with phosphate most likely derived from the body itself.

Introduction

Ichthyosaurs were marine pursuit predators that lived from the Early Triassic to the middle Cretaceous (Fischer et al., 2016). Their rich fossil record of skeletal elements and teeth are ideal for analyses of ecological interactions and evolutionary patterns in the marine realm (spanning approx. 160 million years), and display many examples of convergent evolution to cetaceans, e.g., in body shape and swimming performance (Cozzi, Huggenberger & Oelschläger, 2017; Lindgren et al., 2018). To fully understand their adaptations, fossilized soft tissue is crucial, but presently, the nature and diagenetic processes affecting soft tissue preservation in ichthyosaurs are not well understood (Lindgren et al., 2018).

Soft tissue in ichthyosaur fossils has been observed for almost two centuries (e.g., Fraas, 1888; Lydekker, 1889; Martill, 1987; Martill, 1995; Martin, Frey & Riess, 1986; Owen, 1841; Whitear, 1956), see summary in Martill (1993). The vast majority are specimens from the Lower Jurassic (Hettangian to Toarcian) and some from the Middle Jurassic (Callovian) of Europe. The most famous of these are Stenopterygius specimens from the Toarcian Posidonia Shale of Holzmaden in Baden Württenburg, Germany. Preserved body outlines show that ichthyosaurs had a dorsal fin and a vertically oriented caudal fin (Böttcher, 1990; Eriksson et al., 2022; Huene, 1922; Lindgren et al., 2018; Martill, 1993; Maxwell, 2012a; Maxwell, 2012b; Renesto et al., 2020). Recent studies have revealed the structure and color of ichthyosaur skin in a few key specimens from the Early Jurassic and shown that some ichthyosaurs had blubber (Lindgren et al., 2014; Lindgren et al., 2018). However, for later ichthyosaurs (Middle Jurassic-Cretaceous), records of soft tissue are few, including an Ophthalmosaurus specimen from the Middle Jurassic of England (Martill, 1987), a Thalassodraco and an undescribed ophthalmosaurid specimen from the Upper Jurassic of England (Jacobs & Martill, 2020; K1747, D. Lomax, 2022, personalcommunication) a Maiaspondylus specimen from the Lower Cretaceous of Canada (Maxwell & Caldwell, 2006) and ichthyosaurs from the Solnhofen Plattenkalk.

The Solnhofen Plattenkalk (Fig. 1) is a Late Jurassic Konservat-Lagerstätte preserving many vertebrates and invertebrates with soft tissue, most famously Archaeopteryx lithographica (Rauhut, Foth & Tischlinger, 2018). Ichthyosaur fossils were first found in the Solnhofen deposits in 1852 but are uncommon (Bardet & Fernández, 2000; Barthel, Swinburne & Morris, 1990; Meyer, 1863; Wagner, 1853). Examples of Solnhofen ichthyosaurs with preserved soft tissue have been described by Bauer (1898), and in two specimens of Aegirosaurus leptospondylus (Bardet & Fernández, 2000).

Figure 1 Map showing the Solnhofen area in Germany, and the Eichstätt-Blumenberg visitor quarry (latitude 48.9°N, longitude 11°E) where the ichthyosaur specimens were found.

Modified from Munnecke, Westphal & Kölbl-Ebert (2008).

Here we describe two ichthyosaur specimens with soft tissue from the Solnhofen area from the collection of Bishops Seminar Eichstätt, housed in the Jura-Museum Eichstätt (Figs. 2 and 3). JME-SOS-08369 is a nearly complete skeleton that was recently excavated and prepared, whereas JME-SOS2183 is a tail that was discovered in 1926. None of them have been formally described previously. In addition, we provide a first analysis of the phosphatized soft tissue in the nearly complete specimen. Phosphatization of soft tissue is common in Solnhofen, and fluorapatite, especially in the form of francolite, typical for preservation of muscle fibers (Barthel, Swinburne & Morris, 1990). Phosphatization of tissues normally occur within the sediment (Martill, 1993). In Solnhofen, fully articulated skeletons with mineralized soft tissue preservation are hypothesized to have been buried within a few weeks, such as due to sediments transported into the basin during storm events (Kemp & Trueman, 2003). Together with a quick re-establishment of the pycnocline after storm events with hypersaline, oxygen-poor to anoxic bottom waters, these conditions make the exceptional preservation possible (Pan et al., 2019).

Figure 2 Late Jurassic Aegirosaurus sp. (JME-SOS-08369) from Eichstätt-Blumenberg, Germany.

(A) in normal light. (B) composite picture in UV light indicating position of pictures in Fig. 7. (C) interpretative drawing. L, left; R, right. Type I and II soft tissue marked. Scale bar = 20 cm.

Figure 3 Late Jurassic ichthyosaur tail (JME-SOS2183) from Eichstätt-Blumenberg, Germany.

(A) Regular light. (B) Ultraviolet light. (C) Interpretative drawing. Scale bar = 10 cm.

The phosphorus (P) is commonly interpreted to originate from an external source (Wilby & Briggs, 1997), but in this contribution, we discuss whether it could instead be derived from the body of the animal itself. As previous studies on ophthalmosaurid specimens with soft tissue were only executed using regular light without mineralogical or elemental analyses (Bardet & Fernández, 2000; Maxwell & Caldwell, 2006), this is the first geochemical and UV analysis for this clade.

The apatite supergroup of minerals contains more than 40 different mineral species, but all have the same basic crystal structure. The flexibility of the apatite structure means that it can incorporate many elements into the general formula M12M23(TO4)3X; where M are large cations, like Ca2+, Sr2+, Pb2+, Ce3+, Na1+; T are high valance cations like P5+, As5+, V5+, Si4+, S6+ and X are anions like F−, Cl−, (OH)− or O2−. Hydroxylapatite, ideally Ca5(PO4)3(OH), is the mineral forming bones and teeth. Although mineral formulas are often given as ideal endmembers, like above, apatite typically incorporates other elements in significant amounts, without resulting in formation of a new mineral species. The substitution mechanisms can be homovalent, e.g., F− substituting (OH)−, or heterovalent. To maintain the overall charge of the structure, the heterovalent substitutions typically involves multiple sites in the structure, which is known as coupled substitution. Typical examples of coupled substitutions in apatite are: Ce3+ + Na+ ↔ 2Ca2+; Ce3+ + O2− ↔ Ca2++ F− or S6+ + Na1+ ↔ P5+ + Ca2+. For a review of chemical variation and substitution mechanisms in apatite see Pan & Fleet (2002). The term francolite is often used for sedimentary calcium phosphate, but is not a valid mineral species name, rather a carbonate-rich apatite with a varied composition between fluorapatite and hydroxylapatite (Knudsen & Gunter, 2002; McArthur, 1985; Pasero et al., 2010). Despite not being a valid mineral species, we prefer to use the term francolite here for consistency with other papers in the field.

Geological setting

The ichthyosaur specimens examined here originate from the locality Eichstätt-Blumenberg of Bavaria, southeastern Germany. The strata are the early Tithonian (Late Jurassic) Eichstätt Plattenkalk, of the Kimmeridgian to early Tithonian Solnhofen Archipelago (Fig. 1). At the time of deposition, the area was situated around 34 degrees north (Munnecke, Westphal & Kölbl-Ebert, 2008), at the northern margin of the Tethys Ocean. The Plattenkalk deposits are carbonate successions of finely laminated limestone deposited in a series of subtropical, shallow to deeper basins within a huge carbonate platform (Kölbl-Ebert, Röper & Leinfelder, 2005). The carbonate platform was formed largely by sponge-microbial reefs and accumulated carbonate sands, and locally by small coral reefs (Koch, Senowbari-Daryan & Strauss, 1994; Munnecke, Westphal & Kölbl-Ebert, 2008). The bottom waters of the basins within the platform were primarily stagnant, and in some cases possibly characterized by an elevated salinity and microbial films leading to Plattenkalk deposition, which might be the cause of the excellent fossil preservation (Barthel, 1970; Gäb et al., 2020; Munnecke, Westphal & Kölbl-Ebert, 2008).

Materials and Methods

The nearly complete ichthyosaur skeleton (JME-SOS-08369) (Fig. 2) was excavated in 2009 and prepared at the Jura-Museum Eichstätt (Radecker, 2014). It was preserved on four slabs and prepared from the field-up side, exposing the specimen in oblique view, showing partly the ventral, partly the right-lateral side. Because a major part of the preparation was done under ultraviolet (UV) light to see details, non-fluorescent and UV stable glue and hardener were used (polyvinylbutyral and a non-reflecting hardener), except on the tail slab, where epoxy was used (Radecker, 2014). For a discussion of use of UV in observing fossils in lithographic limestones see Haug et al. (2009).

The second specimen (JME-SOS2183) (Fig. 3) is an ichthyosaur tail with a soft tissue outline, preserved in lateral view. It was found by Karl W. (last name not known) in Blumenberg in 1926 and has at some point been stabilized by paraloid or an equivalent consolidant (C. Ifrim, 2021, personal communication). It was pictured, but not formally described, by Martill (1993).

The skeletal descriptions and measurements are based on personal inspection by LLD, JHH and MKE in 2017. The specimens were photographed while mounted, in exhibition light, in low angle regular light, and in UV light, with a Fujifilm ×100F camera (ISO 800 – aperture 1/8). A hand-held lamp produced low angle light illumination, and a UV lamp with a wavelength 366 nm was used. Observations derived during the preparation process were contributed by Radecker (2014; 2017, personal communication). Skeletal comparisons to Aegirosaurus leptospondylus are based on personal inspection of a referred specimen (SNSB-BSPG 1954 I 608) by LLD (May 2015). The neotype of Aegirosaurus leptospondylus is housed in a private collection and was not studied.

The nearly complete specimen JME-SOS-08369 preserves fossilized soft tissue of several types. Three samples (JME-SOS-08369-B1, -B2 and -B3, Fig. 4) from the anterior portion of the tail and the surrounding sedimentary matrix were set aside during preparation and form the basis for this first investigation of the soft tissue. To assess the nature of the soft tissue, XRD and SEM-EDS analyses were performed on the untreated samples of soft tissue. For the XRD analyses, small fragments (max diameter 100–200 µm) were removed from the three samples and mounted with oil in a cryoloop. The analyses were performed in Gandolfi-mode on a Rigaku Synergy-S diffractometer housed at the Natural History Museum in Oslo, using CuKα-radiation (50 kV and 1 mA). Diffraction data were collected and processed using the CrysAlisPro software, and diagram matching was performed with Bruker’s EVA program with PDF-4 as the database. Based on the XRD analyses, two samples (JME-SOS-08369-B1 and -B2) were chosen for detailed semi-quantitative elemental analyses because they contain apatite in addition to matrix, whereas the third sample (JME-SOS-08369-B3) only contains matrix. The analyses were executed with a Hitachi S-3600N SEM equipped with a Bruker XFlash 5030 energy dispersive detector (EDS) and performed directly on uncoated samples under low vacuum (~10 Pa) with an acceleration voltage of 15 kV and 15 mm working distance. In addition to point analyses, element distribution maps were generated for selected areas of interest. To evaluate the possibility of soft tissue as the phosphorous source for phosphatization, a mass balance was performed. Its constraints were based on cetaceans with similar body shape and size as JME-SOS-08369. For more detailed description of the protocol, see Supplemental Materials. Soft tissue samples from the isolated tail specimen JME-SOS2183 were not analyzed in the present study.

Figure 4 Samples analyzed from Late Jurassic Aegirosaurus sp. (JME-SOS-08369), for possible soft tissue.

Sample JME-SOS-08369-B1 in (A) normal light and (B), UV light. Sample JME-SOS-08369-B2 in (C), normal light and (D), UV light. Sample JME-SOS-08369-B3 in (E), normal light and (F), UV light. Arrows in (B) and (D) on areas where soft tissue epimorphs (apatite) were preserved.

Results

Systematic paleontology

Ichthyosauria de Blainville, 1835

Parvipelvia Motani, 1999

Thunnosauria Motani, 1999

Ophthalmosauridae Baur, 1887

Aegirosaurus Bardet & Fernández, 2000

Aegirosaurus sp.

Referred specimen: Nearly complete skeleton JME-SOS-08369 (Figs. 2, 5, 6, 7, Table 1). The specimen is housed in the Jura-Museum Eichstätt, Eichstätt, Germany.

Figure 5 Skull from Late Jurassic Aegirosaurus sp. (JME-SOS-08369) from Eichstätt-Blumenberg, Germany in (A), interpretative drawing and (B), photograph.

Abbreviations: an, angular; ar, articular; dt, dentary; j, jugal; mx, maxilla; pmax, premaxilla; po, postorbital; sa, surangular; sp, splenial; st, supratemporal. Scale bar = 5 cm.

Figure 6 Selected skeletal elements from Late Jurassic Aegirosaurus sp. (JME-SOS-08369) from Eichstätt-Blumenberg, Germany.

(A) Gastral basket in UV light. Scale bar = 50 mm. (B) Left forefin elements in normal light. Scale bar = 10 mm (C) Ilium in UV light. Scale bar = 10 mm

Figure 7 Selected pictures of Late Jurassic Aegirosaurus sp. (JME-SOS-08369) from Eichstätt-Blumenberg, Germany.

(A) Posterior portion of skull in UV light in posterior-lateral view with type I and II soft tissue. (B) Dorsal outline of specimen with type I soft tissue in UV light. (C) Anterior portion of the abdomen in regular light. (D) Posterior portion of the abdomen in UV light with type II soft tissue. (E) Posterior portion of vertebral column and caudal fin in regular light with type I and II soft tissue. Rectangle shows the area where samples for XRD and EDS analyses were taken. (F) Caudal fin in UV light with type I and II soft tissue. All scale bars 5 cm.

Table 1 Measurements for ichthyosaur specimens from Solnhofen.

All measurements given in millimetres.

	JME-SOS8369	JME-SOS2183	
Total length snout to tail	1,610		
Anteroposterior skull length	440		
Orbit	H = 80, L = 100		
Anteroposterior maxilla length	134		
Premaxilla length anterior to maxilla	205		
Vertebrae directly anterior to caudal fin (n = 3)		H = 14–18, L = 7–8	
Vertebrae in caudal fin		H = 5, L = 5	
Dorsoventral height of peduncle of caudal fin	84	64	
Dorsoventral height of caudal fin (max)		520	

Locality: The visitor quarry in Eichstätt-Blumenberg.

Age: Early Tithonian (eigeltingense horizon), Late Jurassic.

Taxonomic assignment

The nearly complete specimen JME-SOS-08369 can be referred to Ophthalmosauridae with confidence, based on the extensive lateral exposure of the angular (Fernández & Campos, 2015; Moon, 2017). When compared to Aegirosaurus leptospondylus, JME-SOS-08369 shares the following diagnostic characters (Bardet & Fernández, 2000): a snout not markedly demarcated from the skull, delicate and small, densely packed teeth strongly anchored in a dental groove, an enamel crown that is minutely ridged or smooth, a medium sized orbit (orbital ratio 0.22, in between the two Aegirosaurus specimens), a jugal extending anteriorly to the orbit, and a long and very slender rostrum. The skull and total body length fall within the original description of the species (Bardet & Fernández, 2000). However, the status for more than half of the diagnostic characters cannot be accessed in JME-SOS-08369, including the shape and size of the naris, orbit and sclerotic plates, traits in the postorbital/cheek region, as well as all diagnostic postcranial characters. JME-SOS-08369 differs from Aegirosaurus leptospondylus in one aspect: it seems to have a maxilla with relatively long lateral exposure, whereas in Aegirosaurus leptospondylus the exposure is small (Bardet & Fernández, 2000) (L. L. Delsett, 2015, personal observation, SNSB-BSPG 1954 I 608). In conclusion, based on the available data, we refer JME-SOS-08369 to Aegirosaurus sp, pending further review.

Taphonomy JME-SOS-08369

The specimen (Fig. 2) is articulated and preserved in its entire length, missing only the dorsal portion of the caudal fin. Its total length as preserved is 1,610 mm from the tip of the snout to the tip of the ventral caudal fin lobe measured in a straight line. During or after death, the animal landed dorsolaterally on the sea floor. Preservation of the skeletal elements is rather poor. The skull (Fig. 5) is compressed, and is seen from the right side Its total anteroposterior length is 440 mm. The jaw consists of the tooth-bearing right premaxilla and maxilla seen in lateral view, as well as the right dentary, surangular, splenial, and angular. Not all sutures in the skull are well preserved, and the accurate shape of certain elements, such as the maxilla, is not entirely clear. The posterior portion of the left lower jaw elements are preserved and seen in medial view, and a small portion of the right splenial is preserved in lateral view. The lower jaw extends slightly beyond the upper jaw anteriorly, but it is not known whether this is an actual underbite or a taphonomical artifact. Anterior to the orbit and in the posteriormost portion, the skull has suffered considerable distortions, and only a partial postorbital is preserved. The preserved soft tissue is described below.

Description of JME-SOS-08369 and comparison to species within Ophthalmosauridae

Premaxilla

The premaxilla is slender, laterally convex, and increases in dorsoventral height posteriorly. Several anteroposteriorly elongated foramina are situated in the anteriormost portion of the element. Posterior to them is the fossa premaxillaris which extends to the posterior end of the premaxilla, as in Palvennia hoybergeti and Ophthalmosaurus icenicus (Delsett et al., 2018; Moon & Kirton, 2016). In contrast, Sveltonectes insolitus bears a continuous groove (Fischer et al., 2011).

Maxilla

In lateral view, the maxilla extends significantly anterior to the anteriormost border of the naris and possesses a processus narialis. The maxilla is exposed for 40% of the total length of the premaxilla and maxilla, similar to the exposure in Undorosaurus? kristiansenae (38%) and Janusaurus lundi, and in contrast to Aegirosaurus leptospondylus and Sveltonectes insolitus, which have small maxillae relative to the other elements (Bardet & Fernández, 2000; Druckenmiller et al., 2012; Fischer et al., 2011; Roberts et al., 2014). The maxilla is dorsoventrally taller than in Undorosaurus? kristiansenae (Druckenmiller et al., 2012).

Jugal

Anteriorly, the jugal only slightly exceeds the orbital margin, as is also the case for Aegirosaurus leptospondylus (Bardet & Fernández, 2000) and Ophthalmosaurus icenicus (Moon & Kirton, 2016). Its anterior portion is of similar height throughout, as in Athabascasaurus bitumineus, but is not as slender as in Leninia stellans (Druckenmiller & Maxwell, 2010; Fischer et al., 2013). The anterior portion of the jugal is more strongly curved than the jugal of Undorosaurus? kristiansenae and Leninia stellans, in which it is nearly straight (Druckenmiller et al., 2012; Fischer et al., 2013). The curvature of the element is more similar to Palvennia hoybergeti, but with a shorter anterior reach (Delsett et al., 2018).

Angular

The angular is extensively exposed laterally as in Aegirosaurus leptospondylus and most other ophthalmosaurids (Fernández & Campos, 2015), in contrast to the short anterior reach in Palvennia hoybergeti (Delsett et al., 2018). The element reaches approximately as far anteriorly as the surangular, which is slightly anterior to the posterior portion of the premaxilla. The angular increases in dorsoventral height in the posteriormost portion, ventral to the orbit.

Dentary

The dentary increases only slightly in dorsoventral height posteriorly, resulting in a relatively slender element, which is more similar to Aegirosaurus leptospondylus than to Palvennia hoybergeti and Ophthalmosaurus icenicus (Bardet & Fernández, 2000; Delsett et al., 2018; Moon & Kirton, 2016).

Dentition

On the right side, 18 teeth are preserved in the maxilla, 17 in the premaxilla and approximately 20 in the dentary, which means that the specimen has more maxillary teeth than reported for Aegirosaurus leptospondylus (Bardet & Fernández, 2000). The teeth sit closely together in a groove that does not seem to be partitioned into alveoli. The teeth are small, straight, and gracile, more similar to Keilhauia nui and Aegirosaurus leptospondylus (Bardet & Fernández, 2000; Delsett et al., 2017) (SNSB-BSPG 1954 I 608, L. L. Delsett, 2015, personal observation) than to Ophthalmosaurus, Undorosaurus? and Brachypterygius extremus (Druckenmiller et al., 2012; Kirton, 1983; Moon & Kirton, 2016). The margin of the enamel is straight, and the enamel does not have deep ridges.

Pectoral girdle

Three incomplete, slender elements are interpreted as the remains of one or two clavicles, and possibly the interclavicle (Fig. 2). A larger, flat element is interpreted as the coracoid, but information on notches and facets are unavailable.

Humerus

Two partial humeri (Fig. 2) with associated distal elements (Fig. 6B) are preserved. The right humerus is broken and only visible in cross-section. Six forelimb elements with unknown identity are preserved on top of, and in immediate proximity to the humerus, and eleven elements from the left forelimb are preserved partly disarticulated close to the left humerus. All elements are rounded in dorsal or ventral view. Based on their size and position, it is assumed that the elements originate from the proximal portion of the limb. If this is correct, it means that they differ from Aegirosaurus leptospondylus (Bardet & Fernández, 2000) (SNSB-BSPG 1954 I 608, L. L. Delsett, 2015, personal observation), where the distal elements on the forelimb are almost rectangular, except for the distalmost third of the limb. Rectangular forelimb elements are also found in Platypterygius australis, Platypterygius hercynicus, Undorosaurus? kristiansenae and Sveltonectes insolitus, whereas Ophthalmosaurus icenicus and Palvennia hoybergeti possess oval and circular elements (Delsett et al., 2018; Druckenmiller et al., 2012; Fischer et al., 2011; Kolb & Sander, 2009; Moon & Kirton, 2016; Zammit, Norris & Kear, 2010).

Vertebral column

Most of the vertebral column is covered in fossilized soft tissue, and the centra are flattened and partly decomposed (Figs. 2, 7B, 7E). The best preserved are the posterior dorsal and the caudal vertebrae. In JME-SOS-08369, the vertebrae are larger anterior to the tail bend than posterior to it. A complete series of postflexural vertebrae is preserved, with approximately 70 vertebrae. The number of postflexural centra and their shape is similar to the isolated tail JME-SOS2183 described below. Two neural arches are interpreted as those of the atlas-axis and cervical vertebra 3.

The ribs are preserved articulated (Fig. 2). Most ribs do not preserve the proximal portion, but where available, they are bicipital. They possess longitudinal grooves along the entire length, giving a figure eight cross section, as is found in the majority of ophthalmosaurids, but different from Acamptonectes densus, Mollesaurus periallus, and three specimens (PMO 222.667, PMO 224.252, PMO 222.670) from Spitsbergen (Delsett et al., 2017; Delsett et al., 2019; Fernández, 1999; Fischer et al., 2012).

The abdomen (Figs. 6A; 7C, 7D) preserves the remains of an almost complete set of gastralia close to their position in life, probably from the left side, directed slightly posteriorly. Gastralia from the right side are situated on top of the ribs, perpendicular to the direction of the ribs. Their anteroposterior width is less than 5 mm, which is less than half the width of the ribs, and the thickness is uniform throughout. Gastralia in ophthalmosaurids are described in Ophthalmosaurus icenicus and specimens from the Slottsmøya Member of Spitsbergen (Delsett et al., 2018; Delsett et al., 2017; Moon & Kirton, 2016; Roberts et al., 2014). In all of these, the gastralia are circular to subcircular in cross-section and with a smaller diameter compared to the thoracic ribs, as in the specimen described here. The preservation of a nearly complete gastral basket in JME-SOS-08369 is unique among ophthalmosaurids.

Pelvic girdle

Two ilia (Figs. 2, 6C) are preserved, and a few elements that might originate from the hindlimbs are scattered nearby. The ilium is approximately 40 mm in dorsoventral height, slender and straight. Aegirosaurus leptospondylus and Ophthalmosaurus icenicus resemble this specimen in having ilia with a similar anteroposterior width overall, but their ilia are more curved (Bardet & Fernández, 2000; Moon & Kirton, 2016). In contrast, the ilia of Keilhauia nui, Athabascasaurus bitumineus, and Undorosaurus? kristiansenae all have one anteroposteriorly expanded end (Delsett et al., 2017).

Ichthyosauria de Blainville, 1835

Referred specimen JME-SOS2183 (Fig. 3, Table 1). The specimen is housed in the Jura-Museum Eichstätt, Eichstätt, Germany.

Locality The Eichstätt-Blumenberg area. The exact location is not known.

Age early Tithonian, Late Jurassic.

Taxonomic assignment

JME-SOS2183 (Fig. 3) only consists of a tail. There are no diagnostic characters in the tail for Ophthalmosauridae, and the specimen is thus referred to Ichthyosauria indet. Based on its strong similarity to other, more complete specimens from Solnhofen, it most likely belongs in Ophthalmosauridae.

Description and comparison JME-SOS2183

The specimen preserves the peduncle, and the entire caudal fin posterior to it (Fig. 3). As the nearly complete specimen JME-SOS-08369, it preserves a soft tissue outline, only described previously in ophthalmosaurids in the neotype of Aegirosaurus leptospondylus and in a specimen lost in WWII (“Tail Munich State Museum” TMSM), both from Solnhofen (Bardet & Fernández, 2000; Bauer, 1898; Merriam, 1908). The soft tissue outline of the caudal fin is better developed in JME-SOS2183 than in JME-SOS-08369 and demonstrates that the tail had the classical lunate shape in lateral view as in other Jurassic ichthyosaurs, such as Stenopterygius specimens from of Holzmaden. As in the nearly complete specimen JME-SOS-08369, the peduncle of JME-SOS2183 is dorsoventrally narrow and widens dorsally and ventrally into the caudal fin. The dorsal lobe tip is dorsoventrally shorter and anteroposteriorly longer than the ventral portion. The tip of the dorsal lobe is also angled more posteriorly compared to the tip of its ventral lobe. This might however be a taphonomic artefact of preservation and/or preparation, as a crack extends through the matrix and fossil at this point.

Vertebral column

Six caudal vertebrae are preserved articulated anterior to the tail bend and are significantly larger (heights 14–18 mm and length 7–8 mm) than those posterior to the bend. The posterior caudal vertebrae are amphicoelous, and equally anteroposteriorly long dorsally and ventrally. The vertebrae forming the tail bend are not preserved.

Approximately 70 small postflexural vertebrae are preserved in articulation. This is similar to the nearly complete specimen JME-SOS-08369 and to the TMSM (Bardet & Fernández, 2000; Bauer, 1898; Merriam, 1908). The neotype of Aegirosaurus leptospondylus preserves the entire tail, with 61 postflexural centra (Bardet & Fernández, 2000). The dorsoventral height and anteroposterior length of all the postflexural vertebrae in JME-SOS2183 are similar, giving a quadratic outline in lateral view (height = 5 mm, length = 5 mm). Consequently, they are anteroposteriorly longer than those anterior to the tail bend, as in the TMSM specimen (Merriam, 1908). The exception are the miniature vertebrae at the very end of the tail. This is similar to some specimens of Ophthalmosaurus icenicus (Moon & Kirton, 2016), but see (Buchholtz, 2001). Height is also slightly larger than length in the postflexural vertebrae of Acamptonectes densus (Fischer et al., 2012: SNHM 1284-R, fig. 8), Sveltonectes insolitus (Fischer et al., 2011), and Arthropterygius (Maxwell, 2010).

Results–characterization of soft tissue

Several different types of soft tissue are preserved in the nearly complete specimen JME-SOS-08369, based on observation in UV, regular and low angle light (Figs. 2, 4, 7, 8). Here, we discuss two types (type I and II, Table 2). In addition, the abdomen and pelvic girdle regions especially preserve additional types of soft tissue that should be considered for analysis. In these regions, soft tissue appears to be preserved with several different colours and textures and might represent muscles, inner organs, connective tissue, or stomach content. A rectangular lump of fossilized soft tissue in the approximate anteroposterior midpoint of the dorsal outline might represent the remnants of a folded dorsal fin (Fig. 7B). No scale impressions are observed in the skin of any of the specimens.

Table 2 Summary of two described soft tissue types observed in the Aegirosaurus sp. specimen JME-SOS8369.

Type	Location	Colour	Appearance	Preservation	Interpretation	
Type I	Outline of specimen; skull, pelvic girdle and tail	Off-white in regular light, yellowish fluorescence	Relatively thick, no scales	Incrustation pseudomorphs, phosphatic	Epidermis and dermis	
Type II	Most places within the specimen	Light yellow	Amorphous	Incrustation pseudomorphs, phosphatic	Decomposed blubber	

Type I is a thick, outer phosphatic layer, preserved with an off-white color, with yellowish fluorescence (Figs. 7A, 7B, 7E, 7F). It can be observed along the dorsal outline of the specimen starting from the posterior end of the skull, posterior to the pelvic girdle, and in the ventral portion of the tail. During preparation, an additional soft tissue type layered with Type I was observed (Radecker, 2014), but this layer is now preserved only in a few smaller patches. Type II is the most common soft tissue preserved in JME-SOS-08369 (Figs. 2, 7A, 7D–7F). It is an amorphous and light yellow substance infilling large parts of the specimen, covering many of the skeletal elements. Both type I and type II soft tissue are preserved as incrustation pseudomorphs (epimorphs), with no sign of subcellular preservation.

The remnants of soft tissue in the isolated tail JME-SOS2183 are preserved around the entire specimen and was likely enhanced by the addition of some sort of stabilizer/lacquer many years ago (A. Radecker, 2017, personal communication). Type I soft tissue preservation as seen on the leading edge of the caudal fin differs from that on the posterior margin, by being thicker and more robust (Figs. 3A, 3B). A difference in colour might indicate a different tissue composition On the dorsal leading edge, this tissue is segmented.

XRD analyses of material in two samples (JME-SOS-08369-B1 and B2, Fig. 4) that appear off-white in regular light and fluorescent light yellow in UV light, match apatite and is likely referable to type I soft tissue. The same color in UV light is present in the isolated tail JME-SOS2183 in the leading edge of the dorsal lobe of the tail, corresponding to the position of the analyzed samples from JME-SOS-08369, probably representing the same type of mineralized soft tissue. The light beige matrix present on all analyzed samples (JME-SOS-08369-B1, B2 and B3) fits calcite, as expected for limestone.

XRD cannot distinguish between the various members of the apatite group, and chemical data is required to confirm the identification. The semi-quantitative analyses (EDS) (Fig. 8) confirm that the matrix is calcite containing minor amounts of Mg and a silicate mineral, but no phosphates. The material identified via XRD as apatite contains P, Ca, O and F as the main constituents, with minor Na and S, and is consistent with fluorapatite. Apart from Ca and O these elements were only observed in the off-white-colored material. The analyzed samples also have some black material, which is found to be iron oxide with minor Mn. The humeri were covered with a late encrustation of iron hydroxide when excavated (Radecker, 2014).

Figure 8 SEM-EDS of Late Jurassic Aegirosaurus sp. (sample JME-SOS-08369-B2) from Eichstätt-Blumenberg, Germany.

(A) SEM back-scatter image where the circle shows the area from which the EDS analysis of (B) is taken. (C)–(F) are element maps corresponding to (A), showing the sharp boundary between the fossil (rich in P in the form of apatite) and the surrounding matrix free of apatite. Scale bar is 100 µm for all images.

To determine whether the phosphorous could result from an internal source, we calculated whether the soft tissues in a similar-sized cetacean could produce a sufficient amount of phosphorous for the observed amount of phosphatized soft tissue in the nearly complete specimen JME-SOS-08369 (Supplemental Information). The calculations on the amount of phosphorous in the soft tissue (blubber, skin, muscles) for an ichthyosaur this size, yield a total of 32 g phosphorus. The density of fluorapatite is 3.2 g/cm3, which means the decomposition of the specimen could create 54 cm3 pure fluorapatite. Based on its outline (Fig. 2), the area of the specimen is 6,560 cm2, which means that if the apatite formed a single solid crystal under the fossil, it could be 0.8 mm thick. The apatite is not a solid crystal, but will form on the particles within the sediment and pores around the specimen. Consequently, the zone containing apatite could be thicker than 0.8 mm just from phosphorus derived from the soft tissue, even without considering the phosphate derived from the later decomposition of the skeleton. It should be highlighted that by using ideal fluorapatite in our calculation, the amount of apatite that can be formed is a minimum value. For example, the incorporation of carbonate groups for phosphate groups in the apatite structure, would result in more apatite formed, as the content of P in carbonate apatite is smaller than in fluorapatite. That is, the more carbonate that substitutes for phosphorous in the apatite, the more apatite can be formed from the same amount of P. The incorporation of carbonate for phosphorous in apatite also leads to a reduction in density, which means that for the same weight of apatite formed, carbonate-rich apatite will have a larger volume than the same weight of fluorapatite. Consequently, with carbonates substituting for some P in the apatite structure, two effects will increase the volume of apatite that forms.

Discussion

Soft tissue

Type I soft tissue, found along the margins of the body of the nearly complete specimen JME-SOS-08369 and surrounding the tail in JME-SOS2183, is interpreted as smooth epidermis and dermis, possibly including connective tissue in the tail. Previous analysis found that the dermis and epidermis were multi-layered, as in modern tetrapods (Lindgren et al., 2018), which fits with the observation during preparation of at least two layers of soft tissue (Radecker, 2014).

In the dorsal portion of the tail of the SM neotype of Aegirosaurus leptospondylus, Bardet & Fernández (2000, p. 508) observed minute, rounded ‘scale structures’ interpreted as scales covering the skin, but did not figure or elaborate on them. This contrasts with the specimens described here, which are in line with other recent studies, where ichthyosaur skin is interpreted to be smooth (Lindgren et al., 2018).

Type II soft tissue resembles mostly the light yellow, amorphous substance also observed by Lindgren et al. (2018) in a Stenopterygius specimen (MH432) and interpreted by them as adipocere from decomposed blubber. Under conditions with little or no oxygen, microorganisms produce polyhydroxy fatty acids from fatty tissues from a surplus of organic compounds, commonly known as adipocere (Schoenen & Schoenen, 2013).

The EDS and XRD analyses (Fig. 8) show that the soft tissue in the tail of the nearly complete specimen JME-SOS-08369 is phosphatized, likely by francolite, as is common in vertebrate soft tissue from Solnhofen (Barthel, Swinburne & Morris, 1990). As most of the fossilized soft tissue in vertebrates from the Solnhofen area preserve subcellular details, Wilby (1993) and Wilby, Briggs & Viohl (1995) suggested a supersaturated (probably sedimentary) source of phosphorus, because of depleted levels of sedimentary phosphate adjacent to the fossils (Briggs, 2003; Briggs & Wilby, 1996). This contrasts with the phosphatization in the Santana Formation (Brazil), which is interpreted as authigenic mineralization due to decomposition of remains containing phosphate (Briggs, 2003; Briggs et al., 1993; Martill, 1988). JME-SOS-08369 shows no sign of subcellular preservation. The phosphate mineralization is instead observed as incrustation pseudomorphs (epimorphs), resulting from a process by which the soft tissue is coated by apatite, and the encased tissue decomposes. The encasing phosphate remains intact as a thin sheet and retains the outer shape of the original tissue.

Source of the phosphorous

The calculations showed that skin, muscular tissue and blubber could provide sufficient P for the formation of a 0.8 mm thick layer of fluorapatite, i.e., sufficient for the observed preservation of mineralized epimorphs of soft tissue. The other main component of apatite is Ca, found in abundance throughout the host rock (Fig. 8). The skeleton is a possible phosphorous source. However, the apatite layer is not concentrated around the skeletal elements, but rather through the entire area covered by the specimen, indicating that the soft tissues are likely a major phosphorous source.

We conclude that an external phosphorous source is not necessary for this type of preservation to occur. In addition to muscles, we argue that the ichthyosaur possessed blubber, based on our observations of Type II soft tissue, in accordance with previous work (Lindgren et al., 2018). Thus, the phosphorus in the mineralized soft tissue is most likely derived from the most local source, the decaying organic matter of the ichthyosaur itself. A local source of phosphorus rather than external is further supported by the detailed investigation of the specimen and the element mapping (Fig. 8), which show phosphorus only in the area where the fossil is preserved. If the phosphorus source was external, it seems unlikely that apatite only formed where the fossil is preserved. However, as analyses of modern whale-falls have shown that there is an exchange of chemical components between decaying carcasses and the surrounding sediments, an external phosphorous source cannot be completely excluded. The observed preservation of the ichthyosaur is possible because of a short post-mortem floatation with rapid burial and/or overgrowth by microbial films.

Phosphatization of soft tissue in a carbonate-dominated environment, as in the Solnhofen area, has been hypothesized to be dependent on an anoxic environment with low pH (Briggs, 2003; Briggs & Wilby, 1996). In larger vertebrate carcasses, another possible scenario is that the skin creates a local closed system and an elevated concentration of phosphorous during decay. This creates a local fall in pH, shifting the equilibrium away from favouring precipitation of carbonate, to francolite (Briggs, 2003; Briggs & Wilby, 1996; Gueriau et al., 2020).

The source of fluorine in francolite can be explained by the bioapatite presence in the animal, as fluoride incorporation is prevalent although typically low ow with higher concentration close to veins and tissue (Li & Pasteris, 2014). Francolite is known to form by precipitation from anoxic porewater, replacement of carbonate, bacterial processes, post-mortem precipitation of francolite in phosphorus-rich bacterial cells; and possibly by precipitation from the water-column at the sediment water interface (McArthur, 1985).

The minor amounts of Na and S in fluorapatite can be explained by the coupled substitution Ca2+ + P5+ → Na+ + S6+, typical for apatite-group minerals, where Na and S originate from the sea water. In addition, iron was observed through SEM-EDS as well as during preparation, where iron oxide covered some of the elements (Radecker, 2014). This fits well with iron-rich bands observed in SEM-EDS, which are not present in the areas with apatite. The iron oxides and hydroxides are also most likely late weathering products, after the deposits became subaerial in the Cretaceous-Paleogene.

Conclusion

Two ichthyosaur specimens from the Solnhofen Archipelago represent rare cases of soft tissue preservation in ophthalmosaurids, including skin, potential blubber, and additional tissue types. The phosphate mineralization is observed as incrustation pseudomorphs (epimorphs) on the soft tissue. SEM and XRD analyses revealed the presence of fluorapatite within the boundary of the fossil. The phosphorus likely originated from the tissues of the ichthyosaur and does not require an external source. This finding is of interest for understanding the taphonomical pathways of large Solnhofen vertebrates. For future work, microscopical and geochemical analysis of different parts of the specimens have the potential to reveal more information Hopefully, additional work on these specimens can be done at a later stage, with targeted analyses of all different soft tissue types.

The specimens also hold the potential for future investigations into locomotion in derived ichthyosaurs, as the entire body, including two tails, preserve soft tissue outline in combination with probable blubber and some of the most complete ophthalmosaurid tail skeletons described.

Supplemental Information

Supplemental Information 1 Skeletal measurements and protocol for calculations.

Click here for additional data file.

We thank A. Radecker warmly for preparing JME-SOS-08369 and for being an important source of knowledge as well as practical support. We also thank M. Ebert for his hospitality during LLD and JHH’s visit to Eichstätt and the Jura-Museum, and O. Rauhut for his help during LLD’s visit to the collection in Munich. We thank C. Ifrim, C. Crook, H. A. Nakrem, D. Larsen, K. L. Voje, G. Gundersen, and N. Castro for administrative and lab support. We thank editors and reviewers for comments that greatly improved the manuscript.

Institutional abbreviations

JME Jura-Museum Eichstätt, Eichstätt, Germany, specimens belong to the collection of the Bishops Seminar Eichstätt

SNSB-BSPG Bayerische Staatssammlung für Paläontologie und Geologie, Munich, Germany

PMO Natural History Museum, University of Oslo, Norway, palaeontological collections

K The Etches Collection, Dorset, UK

Additional Information and Declarations

Competing Interests

Author Contributions

Data Availability

The authors declare that they have no competing interests.

Lene L. Delsett conceived and designed the experiments, performed the experiments, analyzed the data, prepared figures and/or tables, authored or reviewed drafts of the paper, and approved the final draft.

Henrik Friis conceived and designed the experiments, performed the experiments, analyzed the data, prepared figures and/or tables, authored or reviewed drafts of the paper, and approved the final draft.

Martina Kölbl-Ebert conceived and designed the experiments, analyzed the data, authored or reviewed drafts of the paper, and approved the final draft.

Jørn H. Hurum conceived and designed the experiments, performed the experiments, analyzed the data, authored or reviewed drafts of the paper, and approved the final draft.

The following information was supplied regarding data availability:

The data is available in the Supplemental File.

All the specimens are stored in the Bishops Seminar Eichstätt, Eichstätt, Germany: the mounted fossilized body of JME-SOS-08369 combined from four slabs, as well as the samples JME-SOS-08369-B1, -B2 and -B3, and the fossilized tail specimen JME-SOS2183.

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
