# Peer review of "The soft tissue and skeletal anatomy of two Late Jurassic ichthyosaur specimens from the Solnhofen archipelago"

_PeerJ, doi:10.7717/peerj.13173_

## Round 0.1 · original submission · Minor Revisions

I want to thank you for your patience in receiving this decision. I had an unusually difficult time finding qualified people to agree to do a review, with progress further hampered by the holiday season. You should be very pleased with the reviews, as they are overwhelmingly positive, and reflect the high quality of your manuscript. There are a few areas that do need improvement, however, and primarily involve clarification of terminology and interpretation. Therefore I have decided it should be resubmitted with minor revisions.

In addition to the reviewer's comments noted below I would like you to address the following:

1) In the taxonomic identification section, please be more explicit which characters are used to assign the specimen to Aegirosaurus. What characters are considered apomorphic for the species? If all the features mentioned are characters (i.e., used in cladisitic analysis) then please describe them as such. General description of similarity is fine, but if you are assigning a specific taxon, it must be justified with apomorphy-based identification.

2) I agree with Reviewer 3 about clarifying the descriptions of the two types. This is a necessary change to make. A table summarizing the different tissue types (or preservation types?) would help with this and perhaps help set the stage for future work.

3) In lines 407-409, this statement needs further justification before making a functional inference. Is it organized differently (under magnification) or just thicker on the leading edge? Are there other possible explanations for the difference in thickness other than function? An interpretive figure showing this difference is suggested.

4) Lines 504-511 may be better placed earlier in the manuscript, probably in the introduction, as they describe general taphonomic processes common to Solnhofen fossils.

If you have any additional questions, please do not hesitate to contact me. I can be reached directly at [email protected].

Reviewer 1 ·

Basic reporting

This MS on soft tissue preservation on two ichthyosaur specimens was a joy to review! I would like to congratulate the authors especially on the very clear explanation of the chemical terms and processes, which in other taphonomic description is often rather garbled.

Experimental design

sounds fine!

Validity of the findings

The authors are laudably cautious when interpreting their findings, but do not get hung up in unnecessary qualifications that would make the text confusing. Well done!

Additional comments

I noticed a few spelling errors and there are a handful of sentences that could be phrased a bit better. These very minor alterations can be made during the proofs stage.

Line 485 “these tissues” – which?
Line 491 – and?????? You’re implying something? Spell it out for the idiots!
Line 504 ff – you’re hurrying things here a bit. Phrase everything for the idiot who didn’t read the previous sections.

·

Basic reporting

I would like to praise the authors for their excellent study. Well done. I find the work to be well-presented, the material and approach adequately described, and the language clear and unambiguous. However, I feel that the manuscript could be made stronger with a little more description, including a section on taphonomy (focused on the skeletal anatomy), and with additional figures. Regarding these points, I have added more detailed comments into a marked-up version of the pdf.

As a result, I suggest minor revision.

Experimental design

As per above.

Validity of the findings

As per above. Please see additional suggestions on further descriptions and figures in the marked-up pdf.

Additional comments

N/A

Reviewer 3 ·

Basic reporting

1. The designation of different types of soft tissue is slightly unclear. For example, in line 398-399 you describe the types I and II as preservation types, whereas in line 358 you describe them as tissue types. In general, the same biological tissue may have different fossil preservation in different parts of a body, different individuals, different settings, etc. Similarly, different biological tissues may be preserved in the same type of fossil preservation. Please clarify throughout the paper if the tissue types are preservation types, or biological tissue types, or if in this case you interpret the each preservation type to correspond exactly to one biological tissue type.

2. You mention (e.g. in line 386) that several types of soft tissue are preserved, then describe only two of them. Please give a brief overview of the other types as well.

Experimental design

no comment

Validity of the findings

1. In line 494 you interpret the type II soft tissue as remnants of blubber, but I don't think you did any new analyses to support this, or to demonstrate conclusively that tissue type II in the fossil is composed of adipocere (as you mention in line 466 as a hypothesis for this). Please make sure your conclusions about tissue type II reflect that you are not presenting any new conclusive data about the identification of this.

Additional comments

All-in-all this is an excellent paper! I have no expertise to judge the taxonomic components, but the taphonomic components are well-conceived, well-executed, and well-reported.

---

## Round 0.2 · accepted · Accept

I think you addressed all the comments and feedback from myself and the reviewers and see no reason to send this out for review again.